# Childcare attendance and risk of infectious mononucleosis: A population-based Danish cohort study

Klaus Rostgaard[1,2]*, Lone Graff Stensballe[3], Signe Holst Søegaard[1], Mads Kamper-Jørgensen[4], Henrik Hjalgrim[1,2,5,6]

1 Department of Epidemiology Research, Statens Serum Institut, Copenhagen, Denmark, 2 Danish Cancer Society Research Center, Danish Cancer Society, Copenhagen, Denmark, 3 Department of Pediatrics and Adolescent Medicine, Rigshospitalet, Copenhagen University Hospital, Copenhagen, Denmark, 4 Department of Public Health, University of Copenhagen, Copenhagen, Denmark, 5 Department of Hematology, Copenhagen University Hospital, Copenhagen, Denmark, 6 Department of Clinical Medicine, Copenhagen University, Copenhagen, Denmark

* klar@cancer.dk

**Data Availability Statement:** The data used for the analyses are pseudonymized, but otherwise of such detail: sex, exact date of birth, exact dates of

## Abstract

### Background

The risk of infectious mononucleosis (IM) is affected both by crowding and by sibship structure, i.e., number and signed age differential between an index child and a sibling. Siblings provide protection against IM by pre-empting delayed primary Epstein-Barr virus infection with its associated high risk of IM. The association between childcare attendance and risk of IM, on the other hand, has never been studied in a large, well-characterized cohort.

### Methods

Danish children born in July 1992 through 2016 with a completely known simple childcare attendance history before age 1.5 years (n = 908,866) were followed up for a hospital contact with an IM diagnosis at ages 1.5–26 years. Hazard ratios (HRs) of IM for an additional year of exposure were obtained from stratified Cox regression analyses, stratified by sex and year of birth, with age as the underlying time scale, adjusted for sibship structure, and sociodemographic variables including parental ethnicity and maternal age.

### Results

An additional year of exclusively attending a daycare home (max 5 children) yielded HR = 0.90 (95% confidence interval 0.81–1.00), and similarly, each year of exclusively attending a childcare institution (e.g., crèche) yielded HR = 0.94 (0.84–1.06).

### Conclusions

Forwarding enrollment in childcare by a year lowers the risk of IM later in life much less than having an additional sibling of comparable age and has no practical public health implications. We find our results suggestive of a random threshold for successful Epstein-Barr virus

living and moving between municipalities, enrolling in childcare institutions, and similar data for siblings etc that it cannot be considered really de-identified. And any meaningful replication of the study, with the chosen standard methodology would require this level of detail and precision – also to comply with the availability of a "Minimal data set". This type of data is therefore always stored and processed in the facilities of a trusted Danish third party, in this case Sundhedsdatastyrelsen ("The Danish Health Data Authority"). To get access to the relevant data you therefore have to be affiliated with a Danish research institution. For this and other conditions see [https://sundhedsdatastyrelsen.dk/da/english/health_data_and_registers/research_services]. We will produce all relevant code, metadata and exposure data to anyone with a permit to use the data; other data (from the Danish National Patient Register and the Danish Civil Registration System) have to be obtained from Sundhedsdatastyrelsen. We (Klaus Rostgaard, klp@ssi.dk, +45 29176299) are the contact point for queries regarding the project under which this manuscript has been written. An alternative point of contact will be Anders Hviid, Head of Department of Epidemiology Research, aii@ssi.dk, who has never been involved in this project or worked with these data, but has the capacity to provide the same components of the "Minimal data set" to anyone with a permit to use the data. The raw exposure data are stored on a central server managed by a big IT organization, with all that follows of frequent back-ups, back-ups in physically separated locations etc. This is so by research habit and IT policy. The data can ultimately be recreated from raw data at "Statens Arkiver" (the archives of the [Danish] state).

**Funding:** The study was supported by Helsefonden (https://Helsefonden.dk; grant number 19-B-0352 to KR, grant number 17-B-0273 to HH). The funders had no role in the study design, data collection and analysis, decision to publish, or preparation of the manuscript.

**Competing interests:** The authors have declared that no competing interests exist.

infection that is more easily reached by a sibling than the collective of playmates in daycare homes or childcare institutions.

## Introduction

Infection with human herpes virus 4, Epstein-Barr virus (EBV) is ubiquitous, and by adulthood the vast majority of the population is infected [1, 2]. Primary infection with EBV in childhood is usually asymptotic or accompanied by only mild symptoms. Primary EBV infection in adolescence or adulthood, however, is often accompanied by infectious mononucleosis (IM), the estimated proportions in Danish teenagers ranging from 12% to 70% [2]. IM is typically characterized by fever, tonsilitis, lymphadenopathy and fatigue. Primary infection is followed by lifelong, mostly asymptomatic, latent infection of B lymphocytes [3], which occasionally reactivates lytically [4]. Thus, EBV persistence is characterized by the presence of latently infected cells in the blood and the periodical shedding of virus into saliva [5].

The consequences of IM in terms of educational and work-related absence and more rarely neurological, malignant, bone marrow or liver disease are substantial and underappreciated [6]. Together, this has suggested that reducing the IM-associated sequelae through vaccination targeting EBV would be beneficial in some populations and has prompted development of such vaccines [6].

Investigation of the impact of childcare attendance on age-specific IM risk is lacking and may provide useful information for our understanding of the roles of age, mechanism and"dose" of EBV exposure upon IM risk [2, 7, 8]. This information may help inform the design of an EBV vaccination program and guide policy decisions and parental decisions about when and where children should be enrolled in childcare.

Recently, it has been demonstrated that each additional sibling, especially younger siblings, was associated with lower risk of presenting with IM as a teenager [1, 9]. A similar pattern of sibling-induced protection has been observed for EBV- and IM-related diseases like Hodgkin lymphoma [10] and multiple sclerosis [11]. The study by Rostgaard et al. [1] also revealed 1) that the smaller the age difference between the sibling and the index child the lower the IM risk, interpreted as an indirect effect of early pre-emptive EBV infection, and 2) that having siblings age less than four years increased the IM risk acutely, interpreted as a direct effect of EBV transmission at the time. In line with this, follow-up of families with children as IM index cases [9, 12–15] and EBV serotype studies within families [16] both reveal intra-family contagion as an important source of EBV infection.

Childcare attendance might have the same effect on IM risk as having a large sibship, i.e. lowering the risk of IM in teenage years. However, the incidence of hospitalized IM in Denmark was remarkably stable over a long time period during which childcare attendance became the norm [1, 2], suggesting that EBV transmission through exposure to other children in childcare facilities is less effectual or less frequent than through exposure to siblings. The importance of the source of contagion for disease risk has been well-characterized for other diseases [17–21], but has not been established for EBV/IM. Presumably EBV is transmitted through saliva (including sneezing [22]) to the pre-adolescents infected, because most other suggested routes of transmission (transfusion, sex, deep kissing) would seem less relevant for children, while transmission routes of particular relevance to children (breast milk, in utero infection) appear to play a minor role [7, 8, 23–26].

To qualify and quantify the above pre-conceptions we undertook the present study to investigate the impact of childcare attendance and timing of childcare enrollment on the age-specific risk of IM at ages 1.5–26 years, using a Danish childcare database with nation-wide coverage from around the turn of the century [27].

## Materials and methods

The nation-wide Danish Civil registration System (CRS) was implemented on April 1, 1968. All Danish citizens have since been assigned unique identification numbers (the CRS number), by which the CRS continuously monitors individual vital status, emigration status, identity of parents, and residence. The CRS number also allows for identity-secure linkage between health registers [28].

Children born in Denmark in July 1992 through 2016 formed our study base. Using the CRS, we identified their siblings and obtained dates of birth for all these children in order to allow the construction of variables such as age difference between a proband and an exposing sibling.

We collected information about individual childcare attendance at ages 0–6 years from the Childcare Database [27]. This database includes childcare attendance data on children aged 0–6 years living in one of the 98 Danish municipalities (originally 271), with data dating back to 1989. Recently, we updated the database to include childcare attendance data up to and including 2016. The childcare data are collected routinely by the Danish municipalities to organize payment and distribution of places in childcare facilities. We obtained the childcare attendance data from three Danish data management companies: KMD, IST-software, and GK-consult. In addition, archived data were obtained from the Danish National Archives and a local archive at the municipality of Hørsholm.

For each combination of calendar year and municipality we assessed whether childcare registration was complete based on the percentage of children aged 3–5 years enrolled in childcare on January 1st of that year. In the vast majority of municipalities with the highest coverage, the coverage was remarkably similar in any given calendar period, leading to the following criterion. If the registration percentage was below 72 and the calendar year before 2007 or the registration percentage was below 84 and the calendar year after 2006 the municipality was deemed to have incomplete registration for that year. This enabled censoring upon incomplete exposure information. 2007 was the year of the reorganization of the municipalities from 271 into 98.

From the Childcare Database we collected exact dates of enrollment and withdrawal from childcare facilities, and type of childcare facility. The four main types of childcare facilities registered were: daycare home, crèche, kindergarten, and age-integrated. Daycare homes are for max 5 children of the same age, while the other facilities included a mean number of children above 40 [27, 29, 30].

The Danish National Patient Register was established in 1977 and has since recorded 99.9% of all discharges from Danish non-psychiatric hospitals [31]. For each hospitalization, the register contains information on dates of admission and discharge, and discharge diagnoses. In the register, we identified all hospital contacts including outpatient visits containing a discharge diagnosis code B27* (ICD-10) or 075 (ICD-8). ICD-8 codes were used in the register before 1994, ICD-10 codes from 1994 onward.

The study was approved by SSI QA & Compliance (journal no. 20/13012). According to Danish law, no ethical approval nor consent is needed for a purely register-based study such as this.

## Statistical analysis

The cohort of children born in Denmark from July 1992 through 2016 who were available for childcare attendance assessment from time of birth to age 1.5 years was followed for hospitalization for IM from age 1.5 years, until the date of diagnosis of IM, death, emigration, censoring due to unknown exposure status or December 31, 2018, whichever came first. Censoring due to unknown childcare attendance status occurred on the first occasion where the child was living in a municipality with incomplete childcare registration at the time and the child was less than 1.5 years old (end of exposure ascertainment).

Cox regression stratified by sex and year of birth with age as the underlying time-scale was used to model hazard ratios (HRs) and thereby assess the effects of childcare attendance. All analyses were adjusted for time-varying sibship characteristics (number of siblings of a certain age (0,1,2,3 years) and number of siblings with a certain age differential to the index child as in [1]. We also adjusted for some readily available potential socio-demographic confounders: maternal age [32], parental ethnicity [33–37], socio-economic index of the municipality of birth in year of birth [38] and the fraction of childcare exposure time before age 1.5 due to day-care in the municipality of birth in year of birth. In order to capture both genetic and sociodemographic effects of ethnicity in the most effective way we assessed for each parent whether they were born in a Western country, operationalized/approximated as being born outside Europe (excluding Turkey and including USA and Canada). The socio-economic index is a weighted basket of 14 indicators of municipal financial needs and tax incomes, used for redistribution of tax incomes between Danish municipalities in any given fiscal year, and as such is recalculated annually. The socio-economic index is designed to have an average value of 1, and higher values correspond to poorer municipalities. We observed that densely populated/more urban municipalities tended to have the largest part of childcare executed in institutions. The fraction of daycare out of all childcare at age below 1.5 years was designed to remove confounding that would otherwise occur as a consequence of correlation between urbanicity and both outcome and exposure when trying to assess a possible differential effect of exposure in institutions and daycare facilities.

Exposure was defined as the time enrolled in a childcare institution (crèche, kindergarten or age-integrated institution) or a daycare home before age 1.5 years, assuming that the former comprised a more infectious environment than the latter. The age interval of 0–17 months both contained most of the variation in childcare attendance as well as being the period in childhood with the most EBV sero-conversions [2]. The parameter on log-scale corresponding to these predictors is the hazard rate of seroconverting during the first 18 months due to the exposure (and thus be removed from risk of getting IM at a later age).

We only followed up children who had been attending exclusively daycare homes or institution care. Thus, the followed up cohort could be viewed as the observational equivalent of a randomized trial where each child was exposed a random strictly positive amount of time to either daycare home or institution care (childcare in a créche, kindergarten, or similar) before age 1.5 years, but not both.

All analyses were performed using the SAS statistical software package (version 9.4 SAS Institute, Cary, NC, USA). Ninety-five percent confidence intervals (CIs) were based on Wald tests. Fig 1 was prepared using the forestplot package in R, to visually augment Table 1.

## Results

Our sampling frame consisted of the 1,567,388 children born in Denmark in July 1992 through 2016 with a known mother. In total, 1,152,329 of these children had complete childcare exposure information from birth to age 1.5 years and could be followed afterward. Among these,

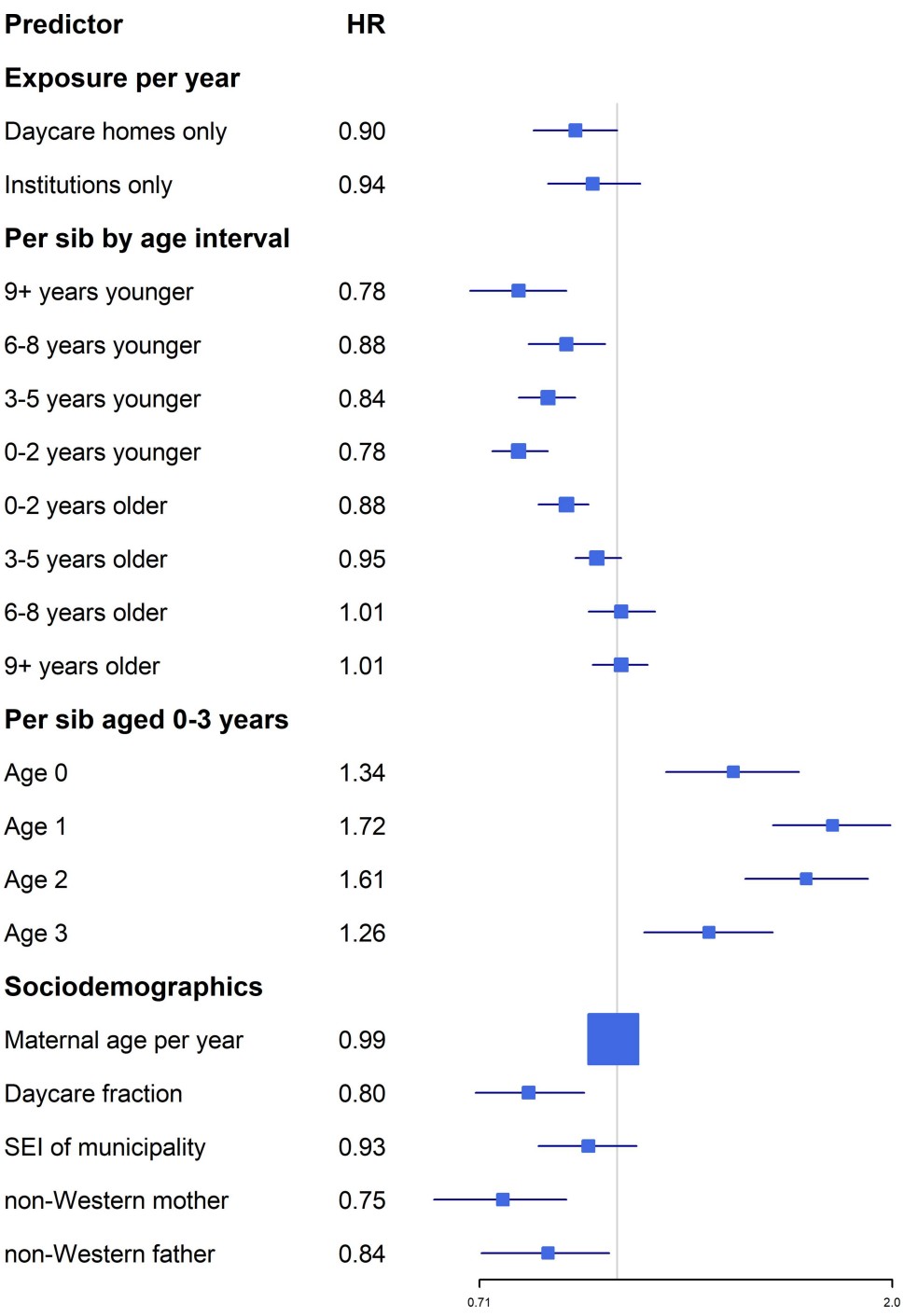

**Fig 1. HRs from model M2B (Table 1) with abbreviated predictor descriptions.**

980,011 individuals had been exposed to childcare before age 1.5 years, of whom 908,866 had been exclusively exposed to either daycare homes or childcare institutions before age 1.5 years. Thus, our study base modelled the 908,866/1,152,329 = 79% of a contemporary Danish birth cohort who were exclusively exposed to either daycare home or childcare institution within the first 1.5 years of life. The distribution of events, follow-up time and contributing persons in

**Table 1. Persons followed up (Persons), person-years (Pyrs) and incidences of IM (Events) by exposure, sex and overall in a cleanly exposed cohort of completely exposure ascertained Danish children born in July 1992 thought 2016.**

| Model | | | | M0 | M1 | M2A | M2B |
|---|---|---|---|---|---|---|---|
| Predictor/Characteristic | Persons | Pyrs | Events | HR (95% CI) | HR (95% CI) | HR (95% CI) | HR (95% CI) |
| All | 908866 | 11327139 | 5068 | | | | |
| Male | 466569 | 5818944 | 2449 | | | | |
| Female | 442297 | 5508195 | 2619 | | | | |
| Exposure before age 1.5 years | | | | | | | |
| (HRs for years attending) | | | | | | | |
| Daycare homes only | 476365 | 6488309 | 2901 | | | 0.96 (0.86–1.06) | 0.90 (0.81–1.00) |
| Institutions only | 432501 | 4838830 | 2167 | | | | 0.94 (0.84–1.06) |
| Daycare home only or institution only | 908866 | 11327139 | 5068 | | 0.92 (0.83–1.01) | 0.94 (0.84–1.06) | |
| Per sib by age interval | | | | | | | |
| At least 9 years younger | 41598 | 303906 | 236 | 0.78 (0.70–0.88) | 0.78 (0.69–0.88) | 0.78 (0.69–0.88) | 0.78 (0.69–0.88) |
| 6–8 years younger | 79061 | 721104 | 410 | 0.88 (0.80–0.97) | 0.88 (0.80–0.97) | 0.88 (0.80–0.97) | 0.88 (0.80–0.97) |
| 3–5 years younger | 216530 | 2256747 | 1097 | 0.84 (0.78–0.90) | 0.84 (0.78–0.90) | 0.84 (0.78–0.90) | 0.84 (0.78–0.90) |
| 0–2 years younger | 215514 | 2517211 | 1035 | 0.78 (0.73–0.84) | 0.78 (0.73–0.84) | 0.78 (0.73–0.84) | 0.78 (0.73–0.84) |
| 0–2 years older | 252963 | 3092306 | 1263 | 0.88 (0.82–0.94) | 0.88 (0.82–0.93) | 0.88 (0.82–0.93) | 0.88 (0.82–0.93) |
| 3–5 years older | 231657 | 2891602 | 1306 | 0.95 (0.90–1.02) | 0.95 (0.90–1.02) | 0.95 (0.90–1.01) | 0.95 (0.90–1.01) |
| 6–8 years older | 101933 | 1268267 | 574 | 1.02 (0.94–1.10) | 1.01 (0.94–1.10) | 1.01 (0.93–1.10) | 1.01 (0.93–1.10) |
| At least 9 years older | 73308 | 919000 | 391 | 1.01 (0.94–1.08) | 1.01 (0.94–1.08) | 1.01 (0.94–1.08) | 1.01 (0.94–1.08) |
| Per sib aged 0–3 years | | | | | | | |
| Age 0 | 445505 | 556626 | 160 | 1.34 (1.13–1.58) | 1.34 (1.13–1.58) | 1.34 (1.13–1.58) | 1.34 (1.13–1.58) |
| Age 1 | 459069 | 554063 | 197 | 1.72 (1.48–1.99) | 1.72 (1.48–2.00) | 1.72 (1.48–1.99) | 1.72 (1.48–1.99) |
| Age 2 | 456643 | 546839 | 185 | 1.61 (1.38–1.88) | 1.61 (1.38–1.88) | 1.61 (1.38–1.88) | 1.61 (1.38–1.88) |
| Age 3 | 516481 | 594121 | 164 | 1.26 (1.07–1.48) | 1.26 (1.07–1.48) | 1.26 (1.07–1.48) | 1.26 (1.07–1.48) |
| Maternal age, per one year increase | 908866 | 11327139 | 5068 | 0.99 (0.98–0.99) | 0.99 (0.98–0.99) | 0.99 (0.98–0.99) | 0.99 (0.98–0.99) |
| Municipality daycare/childcare person-years | 908866 | 11327139 | 5068 | 0.78 (0.69–0.87) | 0.78 (0.70–0.87) | 0.80 (0.70–0.92) | 0.80 (0.70–0.92) |
| Socio-economic index of municipality of birth (mean value is 1 by design, increasing value means more deprived) | 908866 | 11327139 | 5068 | 0.93 (0.82–1.06) | 0.93 (0.82–1.05) | 0.93 (0.82–1.05) | 0.93 (0.82–1.05) |
| Mother not born in Western country | 75468 | 787080 | 228 | 0.75 (0.64–0.88) | 0.75 (0.63–0.88) | 0.75 (0.63–0.88) | 0.75 (0.63–0.88) |
| Father not born in Western country | 70237 | 757535 | 235 | 0.84 (0.72–0.99) | 0.84 (0.71–0.98) | 0.84 (0.71–0.98) | 0.84 (0.71–0.98) |

Hazard ratios with 95% confidence intervals for four models (M0-M2B) of the joint effect of childcare exposure time during the first 1.5 years of life and sibship structure (the "per sib"-predictors), maternal age, and parental birthplace. All models are Cox regressions stratified by sex and year of birth with age as underlying time-scale. Childcare exposure predictors differ between the models: M0 (the baseline) has none, M1 has one overall predictor, M2A and M2B are different parametrizations of the same model that allows for a different effect of exposure to daycare home and institutions; the latter assessing the effect of each on its own, the former assessing the effects as the sum (product of HRs) of an overall effect and a contrast between exposure to daycare homes and institutions.

various strata is illustrated in Table 1. Onset of exposure occurred mainly in a narrow time interval. The 5, 25, 50, 75 and 95% quantiles for the exposure time in years was as follows: exclusively enrollment in daycare home (0.20,0.53,0.67,0.86,1.04), exclusively enrollment in childcare institution (0.17,0.47,0.62,0.75,1.00), and any of these two (0.18,0.50,0.64,0.81,1.03).

The effect of sibship structure (sibling age differentials and having 0-3-year-old sibs) was broadly as expected; i.e., more protection the smaller the difference in age, younger sibs being generally more protective than older sibs and a marked instantaneous effect of having 0-3-year-old sibs on IM risk (see [1]) (Table 1, Fig 1). The estimates were very stable between models, indicating that sibship structure is not confounding the estimation of childcare effects (Table 1).

One additional year of enrollment in a daycare home before age 1.5 years implied a 10% relative reduction in IM risk, while gaining one year in a childcare institution lowered the IM risk by 6% (Table 1). The difference in IM risk due to exposure in daycare homes and exposure in childcare institutions was small and statistically insignificant (Table 1). A one-year increase in exposure time is a lot compared to e.g. an interquartile distance of around 0.3 years for all three types of exposure time. Hence the variation in IM risk explained by childcare exposure would be even smaller.

## Discussion

Childcare attendance has consistently been found to entail a short-term increased risk of childhood infections, all well as affecting the risk of some long-term outcomes, e.g. acute lymphoblastic leukemia, see [29, 30] and references therein. However, the effect sizes we found regarding a phenomenon mainly caused by EBV infection are so small as to have no public health implications. Nevertheless, our study may be informative about how and when infants are infected with EBV—an open and important question in basic IM/EBV epidemiology [7].

The true lasting effect of childcare exposure must be caused by early seroconversion and hence be protective against IM [2, 37]. In our study most children were enrolled in childcare at age 6 to 16 months, where most childhood sero-conversion occurs [2], so that age at enrollment in childcare could make a noticeable lasting difference in IM risk. Taking the observed estimates at face value the effect of bringing forward childcare attendance one year (from age 1.5 to 0.5 years) was much less than the effect of having an additional sibling of roughly the same age, as observed here and in an overlapping study [1]. And there was no suggestion of a trend in the direction of exposure to many children in an institution being more protective against IM than exposure to few children in a daycare facility.

In other childhood infections it has been found that acquiring the infection from a sibling makes the disease course more severe, presumably due to both intensity and duration of exposure diseases [17–21]. This would suggest that "dose" of EBV matters for EBV to succeed in invading and establishing a persistent infection in the host. On the other hand the summary of the few studies on the related topic of EBV infection being accompanied by IM in [2] is that "dose" of EBV does not matter. We also found effect sizes for sibship exposures to be the same whether the outcome was a hospital contact with IM as here, or self-reported IM [1]. Seemingly the simplest way to reconcile these two sets of observations would be to assume the existence of a random threshold for successful EBV infection, such that once this threshold is reached there is no longer correlation between IM outcome and EBV dose; and on the other hand, low dose EBV exposures as expectedly experienced in childcare would typically not suffice to reach the threshold. By implication some children must be exposed repeatedly to EBV before the infection becomes persistent, i.e. each individual has a certain susceptibility to EBV infection [37, 39, 40], and the EBV infection may be eliminated as demonstrated in vitro [41].

From the sibship parameters presented here and in [1] it can be inferred that especially 0–3 year old sibs and those with the smallest difference in age to the index child are the most contagious. Considering the modest effect of childcare attendance in the first 1.5 years of life observed in this study, the question becomes a conundrum: How do (one of) your sibs become infected in the first place at a young age? Three out of 7 families studied in [16] included examples of the same EBV strain in a parent and a child, indicative of transmission from parent to child. In one of the families 4 of 8 children carried the same EBV strain as the mother; presumably some of the children could have become infected through a sibling. Other available studies of EBV-transmission are by design not so informative on this point [12–15]. We also note that apparently parents and other adult contacts of a household increase the shedding of EBV in

the presence of a child with IM [12], raising the perspective that the protective effect of having siblings may actually to some extent be transmitted or mediated by parents and other adults. The stability of the IM occurrence by calendar year [1] also favors parents as main contributors of EBV infection in their children.

A theoretical partial explanation for the predominance of family members as sources of successful EBV infection could be that the EBV they are shedding contain mutant EBV clones specifically suited to escape immune surveillance by the index child's immune system, which to a large extent is shared with other family members. For e.g. HIV it is commonplace [42]. The theoretical possibility has been raised, and intra-host genomic diversity of the EBV is well established, but the extent to which genomic diversification and adaptation occurs in the population of EBV-infected cells in a human host is not currently known [42–44].

It is noteworthy that the difference in effect of exposure to institutional childcare and daycare homes was very small if at all present; and if anything, we would have expected the opposite sign of this difference. Maybe daycare homes resemble a family home more closely in some important way.

### Strengths and weaknesses

The present study has several strengths and weaknesses to consider. We performed a purely register-based study, thus by design avoiding biases regarding recall, participation, outcome and follow-up. Secular trends were tightly adjusted for by using Cox regression stratified by sex and year of birth with age as underlying time scale. Analyses were adjusted thoroughly for sibship structure, which we believe mediates much of the effect of other socio-demographic factors [1]. Analyses were also adjusted for some readily available strong predictors of IM risk: parental ethnicity and maternal age at birth (Table 1). We also adjusted for two predictors characterizing the municipality of the followed up persons at birth: a socio-economic index and the balance between daycare and institution utilization. The former seemed of little importance, the latter turned out to be a strong predictor of IM risk, by being correlated with urbanicity (Table 1). It is not obvious that the ignored socio-demographic factors should be noticeably correlated with age at childcare enrollment/childcare exposure as this is mainly a question about capacity or supply in the municipalities. Effect sizes with hospitalized IM and self-reported IM as outcomes are remarkably similar when considering exposure to other children [1]. Despite accruing more than 5000 outcome events from essentially following up an entire National birth cohort (Table 1), the study ultimately lacks statistical precision. The main cause for this is a combination of small effect sizes (as expected) and little variation in exposure. We do however gain enough information to confidently rank the effect size as numerically smaller than the effect of having a sibling of roughly the same age as the index person. The small effect sizes identified provides a convincing argument why the incidence of IM has remained so stable in Denmark from 1977 to 2008 [1, 2]. Finally, as far as we know, this is the first study ever having childcare as the only or main exposure of interest for an EBV/IM outcome. The variation in context, exposure, design and measurement makes it very difficult to compare and synthesize relevant previous study findings, see [32, 33, 37] for the most recent studies.

We believe that parents' choice between a daycare home and a childcare institution is mainly a matter of what is available and convenient. When children are kept out of childcare for long, we believe this would usually be due to a combination of a stay at home parent and a somehow fragile child,—not primarily a matter of socio-demographic confounding. By design these children were excluded from follow-up after age 1.5 years, and therefore did not influence our results. Children who are weak and have many infections may preferentially attend

daycare homes rather than childcare in an institution, but how this should affect our results is unclear to us. An unknown, but probably small fraction of the followed up children would have attended a daycare home in the informal economy or private childcare or some unregistered type of childcare prior to enrollment in public childcare [27, 29, 30]. If anything, this would bias our results toward the null.

## Conclusion

The risk of IM is affected much less by age at enrollment in childcare within the first 1.5 years of life than by an additional sibling of a comparable age. Biologically, we interpret our results as suggestive of a random threshold for successful EBV infection, that is more easily reached by a sibling than the collective of playmates in a daycare home or a childcare institution.

## Author Contributions

**Data curation:** Signe Holst Søegaard, Mads Kamper-Jørgensen, Henrik Hjalgrim.

**Formal analysis:** Klaus Rostgaard.

**Funding acquisition:** Klaus Rostgaard, Henrik Hjalgrim.

**Investigation:** Klaus Rostgaard, Lone Graff Stensballe, Henrik Hjalgrim.

**Methodology:** Klaus Rostgaard.

**Resources:** Signe Holst Søegaard, Mads Kamper-Jørgensen, Henrik Hjalgrim.

**Writing – original draft:** Klaus Rostgaard, Henrik Hjalgrim.

**Writing – review & editing:** Klaus Rostgaard, Lone Graff Stensballe, Signe Holst Søegaard, Mads Kamper-Jørgensen.

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
