## [Decision Letter · Decision Letter 0]

13 Sep 2021

PONE-D-21-06869

Childcare attendance and risk of infectious mononucleosis: A population-based Danish cohort study

PLOS ONE

Dear Dr. Rostgaard,

Thank you for submitting your manuscript to PLOS ONE. After careful consideration, we feel that it has merit but does not fully meet PLOS ONE’s publication criteria as it currently stands. Therefore, we invite you to submit a revised version of the manuscript that addresses the points raised during the review process.

We look forward to receiving your revised manuscript.

Kind regards,

Gulfaraz Khan, PhD, FRCPath

Academic Editor

PLOS ONE

Journal Requirements:

2. Please clarify whether the Danish Data Protection Agency approved the study, and provide the full name of the IRB mentioned in the Ethics Statement. Moreover, please provide additional information about the patient records used in your retrospective study. Specifically, please ensure that you have discussed whether all data were fully anonymized before you accessed them and/or whether the IRB or ethics committee waived the requirement for informed consent. If patients provided informed written consent to have data from their medical records used in research, please include this information.

Reviewers' comments:

Reviewer's Responses to Questions

**Comments to the Author**

1. Is the manuscript technically sound, and do the data support the conclusions?

Reviewer #1: Yes

Reviewer #2: Yes

Reviewer #3: Yes

2. Has the statistical analysis been performed appropriately and rigorously? 

Reviewer #1: I Don't Know

Reviewer #2: Yes

Reviewer #3: No

3. Have the authors made all data underlying the findings in their manuscript fully available?

Reviewer #1: Yes

Reviewer #2: Yes

Reviewer #3: No

4. Is the manuscript presented in an intelligible fashion and written in standard English?

Reviewer #1: Yes

Reviewer #2: Yes

Reviewer #3: Yes

5. Review Comments to the Author

Reviewer #1: Very nice data and approach to the problem of looking at childcare attendance and IM risk.

As a non-statistician and.or epidemiologist I don't feel qualified to comment on how you prepared your data for analysis. The number of assumptions about what to include and what to exclude is concerning, but you have made attempts to justify why you have done this. However, the data is what it is and there are not many countries that can do this sort of study. Well done on getting this far. Reading your paper has made me think quite deeply about EBV and IM risk.

Reviewer #2: Dear Editor

The manuscript is interesting, trying to answer important clinical points for the first time. In addition, I think it is well written; therefore I suggest it should be accepted for publication.

Reviewer #3: Overall the paper is well written. However, the abstract should nevertheless be revised with respect to language.

The statistical analysis is appropriate but the presentation is difficult to follow. A reason for this is that everything is presented in form of tables. Instead, HRs can be visualized see e.g. a turorial (for R)

https://www.mdpi.com/2504-4990/1/3/58

This will improve the presentation and allows the reader to obtain a better overview. I would present the visualization in addition to the tables.

6. PLOS authors have the option to publish the peer review history of their article (what does this mean?). If published, this will include your full peer review and any attached files.

Reviewer #1: **Yes: **Gavin Giovannoni

Reviewer #2: No

Reviewer #3: No

---

## [Author Response · Author response to Decision Letter 0]

11 Nov 2021

Answers to reviewer comments

Reviewer #1: Very nice data and approach to the problem of looking at childcare attendance and IM risk. 

As a non-statistician and.or epidemiologist I don't feel qualified to comment on how you prepared your data for analysis. The number of assumptions about what to include and what to exclude is concerning, but you have made attempts to justify why you have done this. However, the data is what it is and there are not many countries that can do this sort of study. Well done on getting this far. Reading your paper has made me think quite deeply about EBV and IM risk. 

ANSWER: Thank you very much for these comments.

Reviewer #2: Dear Editor

The manuscript is interesting, trying to answer important clinical points for the first time. In addition, I think it is well written; therefore I suggest it should be accepted for publication.

ANSWER: Thank you very much for these comments.

Reviewer #3: Overall the paper is well written. However, the abstract should nevertheless be revised with respect to language. 

ANSWER: Thank you for pointing out this weakness. The abstract has been revised in 7 places to improve the grammar. Furthermore the sentence ”Hazard ratios…were obtained …. adjusted for sibship structure, parental ethnicity and maternal age” was revised to ”Hazard ratios…were obtained …. adjusted for sibship structure, and sociodemographic variables including parental ethnicity and maternal age” to be thoroughly correct.

The statistical analysis is appropriate but the presentation is difficult to follow. A reason for this is that everything is presented in form of tables. Instead, HRs can be visualized see e.g. a turorial (for R) 

This will improve the presentation and allows the reader to obtain a better overview. I would present the visualization in addition to the tables.

ANSWER: We have added a figure presenting the rightmost column of the table as a forest plot – with abbreviated headings. As the last sentence of ”Statistical methods” we have added: ”Fig 1 was prepared using the forestplot package in R, to visually augment Table 1.” The figure is referenced once in the second paragraph of the results section: ”The effect of sibship structure (sibling age differentials and having 0-3-year-old sibs) was broadly as expected; i.e., more protection the smaller the difference in age, younger sibs being generally more protective than older sibs and a marked instantaneous effect of having 0-3-year-old sibs on IM risk (see [1]) (Table 1, Fig 1).” 

Furthermore we have completed the formatting for PLoS, corrected a few grammatical errors and made some slight additions to a sentence in the introduction: ”Presumably EBV is transmitted through saliva (including sneezing [22]) to the pre-adolescents infected, because most other suggested routes of transmission (transfusion, sex, deep kissing) would seem less relevant for children, while transmission routes of particular relevance to children (breast milk, in utero infection) appear to play a minor role [7,8,23–26]”, and a sentence in the beginning of the Discussion section: ”However, the effect sizes we found regarding a phenomenon mainly caused by EBV infection are so small as to have no public health implications”.

---

## [Editor Report · Decision Letter 1]

9 Dec 2021

Childcare attendance and risk of infectious mononucleosis: A population-based Danish cohort study

PONE-D-21-06869R1

Dear Dr. Rostgaard,

We’re pleased to inform you that your manuscript has been judged scientifically suitable for publication and will be formally accepted for publication once it meets all outstanding technical requirements.

Kind regards,

Gulfaraz Khan, PhD, FRCPath

Academic Editor

PLOS ONE
---

## [Editor Report · Acceptance letter]

13 Dec 2021

PONE-D-21-06869R1 

Childcare attendance and risk of infectious mononucleosis:
A population-based Danish cohort study 

Dear Dr. Rostgaard:

I'm pleased to inform you that your manuscript has been deemed suitable for publication in PLOS ONE. Congratulations! Your manuscript is now with our production department. 

Kind regards, 

on behalf of

Prof Gulfaraz Khan 

Academic Editor

PLOS ONE